# Virtual Reality-Based Cognitive Stimulation on People with Mild to Moderate Dementia due to Alzheimer’s Disease: A Pilot Randomized Controlled Trial

**DOI:** 10.3390/ijerph18105290

**Published:** 2021-05-16

**Authors:** Jorge Oliveira, Pedro Gamito, Teresa Souto, Rita Conde, Maria Ferreira, Tatiana Corotnean, Adriano Fernandes, Henrique Silva, Teresa Neto

**Affiliations:** 1Digital Human-Environment Interaction Lab–HEI-Lab, Lusófona University, 1749-024 Lisboa, Portugal; pedro.gamito@ulusofona.pt (P.G.); p2683@ulp.pt (T.S.); p5025@ulp.pt (R.C.); mjose.ferreira@ulp.pt (M.F.); 2Department of Psychology, Lusófona University, 1749-024 Lisboa, Portugal; tatiana.corotnean@hotmail.com; 3Santa Casa da Misericórdia da Amadora–SCMA, 2610-143 Amadora, Portugal; adrianofernandes@misericordia-amadora.pt (A.F.); henriquesilva@misericordia-amadora.pt (H.S.); teresaneto@misericordia-amadora.pt (T.N.)

**Keywords:** Alzheimer’s disease, aging, computerized cognitive stimulation, dementia, ecological validity, virtual reality

## Abstract

The use of ecologically oriented approaches with virtual reality (VR) depicting instrumental activities of daily living (IADL) is a promising approach for interventions on acquired brain injuries. However, the results of such an approach on dementia caused by Alzheimer’s disease (AD) are still lacking. This research reports on a pilot randomized controlled trial that aimed to explore the effect of a cognitive stimulation reproducing several IADL in VR on people with mild-to-moderate dementia caused by AD. Patients were recruited from residential care homes of Santa Casa da Misericórdia da Amadora (SCMA), which is a relevant nonprofit social and healthcare provider in Portugal. This intervention lasted two months, with a total of 10 sessions (two sessions/week). A neuropsychological assessment was carried out at the baseline and follow-up using established neuropsychological instruments for assessing memory, attention, and executive functions. The sample consisted of 17 patients of both genders randomly assigned to the experimental and control groups. The preliminary results suggested an improvement in overall cognitive function in the experimental group, with an effect size corresponding to a large effect in global cognition, which suggests that this approach is effective for neurocognitive stimulation in older adults with dementia, contributing to maintaining cognitive function in AD.

## 1. Introduction

As the global population ages, the incidence of major neurocognitive disorders (NCDs), such as Alzheimer’s disease (AD), is expected to rise in the following years [1].

According to the World Health Organization [2], AD is the most common form of dementia and may contribute to 60–70% of cases. In 2015, dementia affected 47 million people worldwide (accounting, approximately, for 5% of the older adult population); however, recent reviews have estimated that, globally, nearly 9.9 million people develop dementia each year—this figure translates into one new case every three seconds.

Dementia has a physical, psychological, social, and economic impact not only on people with dementia but also on their caregivers, families, and society. Furthermore, it is one of the major causes of disability and dependency among older people worldwide [3]. Individuals with a diagnosis of dementia will require a diversity of services that include case finding; diagnosis; treatment (including pharmacological and psychosocial); rehabilitation; palliative/end-of-life care; and other support (e.g., home help, transport, food, and the provision of a structured day with meaningful activities) [2].

In Portugal, for every 100 young citizens, there were 129.6 citizens over 65 years old in 2011, but this number has increased dramatically in the last ten years, to 157 in 2020 [4]. By 2050, in Portugal, as well as Japan, Italy, and Spain, dementia will be prevalent in more than one in 25 people, with an estimated prevalence of dementia between 2019 and 2050 of 40.5 per 1000 of the population [5].

### 1.1. Alzheimer’s Disease

Dementia is an umbrella term for several diseases (including AD), usually characterized by a significant decline in one or more cognitive domains (e.g., language, memory, or executive functioning), considering the individual’s age and education. Dementia due to AD is generally characterized by an insidious onset, with a progressive course leading to a deterioration of the functional abilities that eventually culminates in global cognitive impairment and compromised functional independence, bearing a huge personal and societal impact [6].

The cognitive symptoms of AD most commonly include deficits in complex attention, executive functions, perceptual-motor functions, language abilities (i.e., aphasia) and semantic knowledge in learning, and social cognition. However, episodic memory impairment (i.e., amnesia) is usually the earliest and the most pervasive feature of AD [7,8,9]. In the early stages of the disease process, recent episodic memories are the most affected, while memories of the distant past are usually spared. As the disease progresses, all aspects of episodic memory become affected. In contrast, working memory and semantic memory are retained until the later stages of the disease process [7].

Between healthy aging and pathological dementia aging, a predementia territory characterized by Ron Petersen, Glenn Smith, and colleagues from the Mayo Clinic as “mild cognitive impairment” (MCI) [10] may be considered. MCI was defined as a condition in which individuals experience memory loss to a greater extent than one would expect for their age, yet do not meet the criteria for dementia [10]. This is a less severe degree of cognitive impairment when compared to dementia and categorized in the DSM-5 as a Mild Neurocognitive Disorder characterized by a great heterogeneity of deficit profiles. However, the capacity for independence in the Activities of Daily Living and Instrumental Activities of Daily Living is preserved, unlike major NCDs [11].

Given the impact of major NCDs, it is crucial to develop effective, targeted treatments to delay the cognitive and functional declines associated with these diseases and mitigate their devastating personal, family, and social consequences.

Pharmacological interventions, unfortunately, have not shown much efficacy in changing the course of AD dementia [12]. To ensure that people with dementia can maintain a level of functional ability, the need for a more definitive cognitive assessment and effective nonpharmacological intervention for age-related NCDs, including AD, becomes of the utmost relevance, given that no definitive diagnostics or efficacious therapeutics are currently available for these conditions [5].

### 1.2. Cognitive Stimulation and the Use of Virtual Reality

Cognitive stimulation (CS) depends on the regeneration potential of the adult brain and focuses on the overall improvement of the individual’s cognitive and social functioning [13], as he/she is involved in a set of specific and selected activities. Research suggests that CS leads to consistent gains in global cognition, especially in individuals with a diagnosis of mild-to-moderate dementia [14,15]. Systematic reviews on this subject [15,16] have suggested that CS benefits are comparatively greater than those from pharmacological interventions, with additional and significant improvements in the quality of life and well-being, communication, and social interactions.

Regarding CS, in recent years, virtual reality (VR) has been seen as an important resource that can enhance therapeutic gains, namely in patients with major NCDs such as AD [17,18,19]. These studies have indicated that VR has been effective in subjects with cognitive decline, specifically by increasing their ability to perform IADL [20,21].

VR is defined as a technology that digitally provides a three-dimensional environment, allowing people to interact, have different sensory inputs, and change the environment [19]. VR can be immersive or non-immersive, with the level of immersion allowing a sense of presence in the environment [19].

The systematic reviews [19,22] on the effectiveness of interventions using VR resources in major NCDs, mainly with patients with AD, suggest significant improvements in cognition, as well as in well-being, which is encouraging for the possibility of improvements in executive functioning [22].

Despite the diversity of virtual environments that aim to achieve cognitive gains, most of them do not consider multidomain functioning [23,24]. The Systemic Lisbon Battery (SLB) stands out in this approach, used with different groups of participants (substance use disorders, aging, traumatic brain injury, and stroke) to promote cognitive functioning, with cognitive exercises depicting diverse instrumental activities of daily living [24,25,26]. The results of the studies carried out with the SLB support the idea that this is an effective tool for functional cognitive improvement [24], mainly at the level of executive functions. Given the lack of consistent data on VR-based CS in dementia, this study aims to report the impact of CS using the SLB in a group of older adults with mild-to-moderate dementia due to AD.

## 2. Materials and Methods

### 2.1. Trial Design

The design of this study was based on an open-label pilot randomized controlled trial (RCT), as it was not possible to blind the intervention to the patients, therapists, and assessors. This pilot trial aimed to provide preliminary evidence for the feasibility of this intervention to inform subsequent validation trials regarding the effectiveness of this intervention on major NCDs. The trial design consisted of a two-arm parallel design with A and B point assessments. The patients were randomly distributed into experimental and control groups. The experimental group consisted of VR cognitive stimulation at residential care homes for older adults, whereas the control group received treatment-as-usual at care units for older adults.

### 2.2. Recruitment

Recruitment was conducted at residential care homes from Santa Casa da Misericórdia da Amadora (SCMA), which is a large nonprofit health and social care provider in the municipality of Amadora in the Lisbon Metropolitan Area of Portugal. The users from these facilities had different comorbidities but usually comprised older adults above 65 years old with different physical and cognitive morbidities. The population selected for this study comprised older adults with major NCD due to AD identified by a psychologist working at the SCMA premises. The potential participants received written and verbal information about the study for informed consent. The same information was provided to the therapists and families of these patients. After agreeing to participate, the patients were included in a pool of participants for group allocation.

### 2.3. Eligibility Criteria

Psychologists assessed the eligibility criteria in the partner institution of this project. The inclusion criteria were the following: being older adults with AD, fluent in Portuguese, above 65 years old, and keen to participate in the study.

The exclusion criteria were as follows: a history of psychiatric (depression, anxiety, or psychosis) diagnosed disorders and severe language or sensory–motor impairments that prevent participation in these exercises.

### 2.4. Intervention

This program comprised twelve cognitive stimulation sessions delivered by clinical neuropsychologists for 45 min sessions, distributed over two days a week, with a dosage of approximately 9 h, which is considered an average dose length from the data of a recent meta-analysis [27]. The intervention was done using a computerized cognitive stimulation program with non-immersive VR, with exercises depicting the IADL for higher ecological validity. The sessions were structured to aim at different proposed cognitive domains, according to Appendix A. These sessions presented different difficulty levels for progression throughout the intervention. This program has already been studied in other populations for cognitive rehabilitation [28] or cognitive stimulation in healthy aging [24]. The SLB was used in this intervention, which is a computerized version of neuropsychological IADL to assess or promote neuropsychological functioning. The SLB has been in use for over ten years in reference institutions in Portugal. The overall results suggest the SLB as a feasible approach for cognitive intervention, producing higher impacts in general cognitive functions related to and supporting everyday life activities. This trial was the first attempt to study this intervention in patients with major NCDs due to AD. In this study, the SLB was used in a computer with non-immersive VR exposure on a laptop screen of 17 inches.

The SLB comprised nine different tasks distributed in twelve sessions with different difficulty levels. The tasks used for this study were the following: (T1) Morning hygiene, (T2) Shoe closet test, (T3) Wardrobe test, (T4) Memory test, (T5) Virtual kitchen, (T6) TV news, (T7) Grocery store, (T8) Pharmacy, and (T9) Art gallery test. Tasks T1–T6 were conducted inside a virtual apartment, whereas T7–T9 were outdoor tasks where participants needed to navigate to each of the locations in a virtual city (Figure 1).

### 2.5. Outcomes

The outcomes were assessed with the established neuropsychological measures. The sociodemographic data was retrieved from the clinical files of each patient at these institutions. These data were related to gender, age, formal education, civil status, and social support.

#### 2.5.1. Primary Outcomes

The primary outcome was executive functioning, as this intervention focused on ecological validity by using IADL in VR for promoting executive functions. The executive functions were assessed with the Frontal Assessment Battery and the Trail Making Test.

The Frontal Assessment Battery—FAB [29], the Portuguese version [30], is a brief instrument for the assessment of abstraction, mental flexibility, motor programming, sensitivity to interference, inhibitory control, and environmental autonomy, which comprise different domains of executive functioning. Each dimension is scored from 0 to 3, for a maximum score of 18 points.

The Trail Making Test—TMT [31] is a performance test divided into two parts: part A that assesses attention, motor coordination, and information processing speed and part B that assesses working memory, executive functions, and the mental flexibility [32]. For part A, the participants are asked to link numbers in ascending order in a line drawing on a A4 sheet. For part B, the participants are asked to link numbers and letters alternately, in ascending order for the numbers and alphabetical order for the letters. In this study, we assessed the performances using a dichotomized score according to task accomplishment.

#### 2.5.2. Secondary Outcomes

The secondary outcomes were global cognition, functionality, depression, and dementia rating.

Global cognition was assessed with the Mini-Mental State Examination—MMSE [33], which is a brief cognitive screening test used to assess cognitive decline in different populations [34]. The total score is 30 points, where a higher score depicts better cognitive function. The validation for Portugal was conducted by Guerreiro and colleagues [35], whereas Santana and colleagues [34] provided cutoffs of 15 points for illiterate individuals, 22 points for 1–11 years of schooling, and 27 points for 11 years of schooling or above.

The Clock Drawing Test—CDT [36], Portuguese version [37], is a paper and pencil instrument for cognitive screening in dementia. The CDT assesses the visuospatial, constructive, and executive functions. The classification is based on a rating system that assesses the quality of the clock circumference and clock hands.

The Lawton–Brody Instrumental Activities of Daily Living Scale—IADL [38] aims to measure disability in instrumental everyday activities through caregivers’ reports. The instrument allows assessing the level of independence in eight functional domains, such as using the telephone, using transport, shopping, preparing food, housekeeping, laundry, medication, and managing finances [39].

The Geriatric Depression Scale-15—GDS-15 [40] was used as a self-reporting instrument for depression. The translation and adaptation to the Portuguese population was prepared by the Group of Studies on Brain Aging and Dementias in Portugal. The maximum is 15 points, which suggests higher depression levels. The cutoffs are: 0–5 as an absence of depressive symptoms, 6–10 for mild depressive symptoms, and 11–15 for severe depressive symptoms.

The Clinical Dementia Rating—CDR [41] is aimed at evaluating cognition and behavior, as well as the ability to perform activities of daily living. This scale is divided into six cognitive–behavioral categories: memory, orientation, judgment and problem solving, community activities, home activities, and personal care. Each of the categories is classified as 0 (absence), 0.5 (questionable), 1 (mild dementia), 2 (moderate dementia), or 3 (severe dementia), except for the personal care category, which does not have a 0.5 level. The memory category has greater relevance in this test, as this comprises the main symptoms of major NCDs. The final classification of the CDR is obtained from its classification in each individual category [42].

### 2.6. Procedure

This project started with the submission of ethical clearance to the Ethical and Deontological Committee for Scientific Research of the School of Psychology and Life Sciences (CEDIC) of Lusófona University in Lisbon. The project was approved by this ethics committee in October 2018.

The informed consent form (original versions in Portuguese of the informed consent form are available in the Appendix A) was provided to the patients (Annex 1), therapists (Annex 2), and caregivers (Annex 3) in simple and clear language, with information on the objectives of the project, confidentiality, and pseudonymization of the data because of the repeated assessment. After agreeing (patients, caregivers, and professionals) to the informed consent, the patients were screened for eligibility according to the inclusion criteria for participating in this study. The eligibility criteria were screened by a psychologist from SCMA that also created a schedule for each week of the intervention, with the days of the week for the intervention for each patient recruited for the study. The baseline assessment session was done with a psychologist of the institution where the patients received regular follow-up. The baseline assessment was divided into two one-hour sessions. Another technician was responsible for conducting the intervention with the computer for cognitive stimulation in the following 4–6-week period, as stated above under intervention. After the intervention was completed, the same evaluator conducted the endpoint assessment in two different one-hour sessions. The matching between each user was done by a numeric keycode that linked the data with each user. No personal information was recorded. This file was available during the intervention for the clinical staff involved in this study at SCMA.

### 2.7. Statistical Analysis

The statistical procedures were conducted in SPSS Version 25.0. (IBM Corp., Armonk, NY, USA). The data was retrieved from each individual file and imported to SPSS for statistical analysis. As this study involved a mixed design (within and between the group comparisons), these analyses were conducted for repeated measures. Before each analysis, the distribution of the dependent variable was assessed for normality with the Kolmogorov–Smirnov (KS) test. The statistical analyses started with descriptive statistics for sample characterization. The neuropsychological outcomes were assessed with repeated measures ANOVA for 2 × 2 comparisons with normal distribution, whereas the Wilcoxon test for A and B comparisons was done for nonparametric distributions separately for the experimental and control groups. The significant ANOVAs were further explored with simple effects with Bonferroni correction. The alpha level was set at 0.05. The effect size was reported as *r* for the nonparametric statistics being further transformed into *Eta^2^*. The effects from the ANOVAs were reported as *Eta^2^_p_* and *Omega^2^_p_*. The partial omega squared effect was calculated, as this is considered less biased for comparisons across studies than partial eta squared [43].

## 3. Results

### 3.1. Sample Description

The first analysis aimed at describing the sample regarding the sociodemographic data. The initial sample comprised 18 patients, but one patient of the experimental group dropped out from the study. The final sample comprised 17 older adults (12 women) with a mean age of 83 years (*M* = 83.24, *SD* = 5.66). Regarding the level of education, most participants (*n* = 11, 64.7%) had primary education, two (11.8%) had education below the primary level but were literate, and four (23.5%) had education higher than primary schooling. As for civil status, most participants (*n* = 11, 64.7%) were widowed, four (23.5%) were divorced, and two (11.8%) were married, while most of them (*n* = 16; 94.1%) had descendants. This sample was randomly divided into an experimental group (*n* = 10) and a control group (*n* = 7). These groups were compared using Mann–Whitney for age and Fisher’s exact statistics for testing the distribution of the categorical sociodemographic variables between the groups (Table 1). No statistically significant differences between the groups were found for age (experimental group: *M* = 82.60 y, *SD* = 5.42 and control group: *M* = 84.14 y, *SD* = 6.30) or significant associations were found for the remaining sociodemographic variables (*p* > 0.05).

The dementia stage was assessed at the baseline using CDR classification. The frequencies of membership of the CDR categories were calculated and compared statistically between the experimental and control groups. These results indicated that most participants were classified as having moderate dementia (*n* = 8, 47.1%), followed by mild dementia (*n* = 7, 41.2%) and questionable dementia (*n* = 2, 11.8%). The same scoring was used for the individual categories, ranging from 0–3 (severe). Regarding the individual categories of CDR, most patients scored one point (*n* = 9; 52.9%) in memory, two points (*n* = 7; 41.2%) in orientation, two points (*n* = 6; 35.3%) in judgment and problem solving, two points (*n* = 10; 58.8%) in community activities, one point (*n* = 8; 47.1%) in home activities, and most scored one or two points (each *n* = 6; 35.3%) in personal care. The associations between the groups with the individual categories of CDR were tested with Fisher’s exact test. However, they did not reveal statistically significant associations, suggesting that the frequency distributions of the patients among these CDR categories were not different between the experimental vs. control groups. These data are described in Table 2.

### 3.2. Primary Outcomes of Intervention

The primary outcomes were executive functions that were assessed with the FAB and the TMT. The distribution of the total score from the FAB at the baseline and post-treatment assessment followed a normal distribution according to the KS test. Therefore, these differences were tested with a repeated measures ANOVA, with one within-subject factor (baseline vs. post-treatment) and one between-subjects factor (groups: experimental vs. control). The results showed no statistically significant main (*p* > 0.05) or interaction effects (*F*(1, 15) = 2.032; *Eta^2^_p_* = 0.119; *p* = 0.174). The effect size *Omega^2^_p_* = 0.057 was considered small-to-medium, given the benchmark provided by Cohen [44] for equivalent eta squared, with the following cutoffs for small (*Eta^2^* = 0.01), medium (*Eta^2^* = 0.06), and large (*Eta^2^* = 0.14) effects. Table 3 shows an increase in the mean score of the FAB for the post-treatment assessment for the experimental group but without statistical significance, as well as a decrease found in the mean score of the FAB in the control group.

The TMT scores were dichotomized as 0 and 1 for classifying the performances in the test according to not accomplished and accomplished tasks, respectively. Therefore, the comparisons between the baseline and post-treatment assessment were performed with the Sign test to identify positive (change), negative (change), and ties (i.e., no change) between these assessment points. This analysis was done separately for the experimental and control groups. These comparisons were corrected using Bonferroni correction (alpha/2), because two comparisons were conducted for this test: one for the TMT score of part A and the other for the TMT score of part B for each group. The results did not reveal a significant difference in the experimental group for both TMT part A *(Z* = −2.121; *p* = 0.063) and TMT part B (*Z* = −1.342; *p* = 0.50) and in the control group for TMT parts A and B (*Z* = 0.000; *p* > 0.05). Four cases showed positive changes (i.e., improvements from the baseline to post-treatment assessment), and six described ties (i.e., no differences between the baseline and post-treatment) in TMT part A. None of the cases revealed a negative change in the experimental group (i.e., decrease from the baseline to post-treatment assessment). For TMT part B, two positive changes were found, but the number of ties was eight in these cases. None of the cases described a negative change. For the control group, no positive/negative changes were found. All the cases in the control group were ties, revealing no variations in the TMT part B scores in the controls (Table 4). The effect sizes for the nonparametric statistics were reported in the experimental group as *r* = 0.67 for TMT part A and *r* = 0.50 for TMT part B. The effect sizes for the comparisons in the control group were not computed, given the small *Z* statistics for these comparisons. The effect sizes in the experimental group were transformed into *Eta^2^* = 0.44 for TMT part A and *Eta^2^* = 0.26 for TMT part B [45].

### 3.3. Secondary Outcomes of Intervention

The secondary outcomes were based on the global cognition, functionality, depression, and dementia rating scores. Global cognition was assessed using an established cognitive screening test—MMSE, along with the CDT. The distribution of the total score for the MMSE was tested using the KS, which revealed an adjustment to the normal distribution (*p* > 0.05). The repeated measures ANOVA was conducted under the same conditions as for the FAB scores. These results revealed a statistically significant interaction between the factors (*F*(1, 15) = 4.930; *Eta^2^_p_* = 0.247; *p* = 0.042), as depicted in Table 3. The partial omega squared effect was *Omega^2^_p_* = 0.187, which was considered a large effect size [44]. This effect was decomposed using the simple effects analysis, which indicated a significant improvement from the baseline to the post-treatment assessment only in the experimental group (MMSE Δmean = −1.20; *SE* = 0.51; *p* = 0.033). Regarding the group comparisons, no differences were found between the groups at the baseline assessment point. A marginally significant trend was found at the post-treatment assessment for the differences between the experimental vs. control groups (MMSE Δmean = 5.60; *SE* = 3.41; *p* = 0.056).

The CDT was scored on an ordinal scale (0–10). Therefore, these comparisons were performed separately for the experimental vs. control groups with the Wilcoxon Signed Rank test for two related samples. These results did not reveal statistically significant comparisons between the experimental and control groups (*Z* = 0.000; *p* > 0.05). In the experimental group, there was one case with positive ranks (i.e., higher scores at the post-treatment assessment) and one case with negative ranks (i.e., higher scores at the baseline assessment), whereas the remaining cases were ties (i.e., no rank differences between the assessment points). In the control group, all the cases were ties, suggesting no variations in the CDT scores from the baseline to the post-treatment assessment (Table 3). The effect sizes were not computed given the small Z statistics for the comparisons in the experimental and control groups.

Regarding the IADL assessment, the KS revealed an adjustment to a normal distribution (*p* > 0.05). The ANOVA did not reveal statistically significant differences (*F*(1, 15) = 0.015; *Eta^2^_p_* = 0.001; *p* = 0.905) between the baseline and post-treatment assessments (Table 3). The partial omega squared effect was *Omega^2^_p_* < 0.01, suggesting no effects of the intervention on functionality as measured using the IADL.

The GDS-15 was tested for normality using the KS. The KS was only significant for the GDS-15 post-assessment (*p* < 0.05). These comparisons were then conducted using the Wilcoxon Signed Rank test, given that one of the variables was not normally distributed. No significant differences were found between the baseline and post-assessment in either the experimental (*Z* = −1.897; *p* = 0.058) or control groups (*Z* = −0.272; *p* = 0.785). In the experimental group, five cases described a negative change (i.e., higher scores at the post-treatment assessment—in GDS-15, a higher score was indicative of higher depression), one was negative (i.e., a higher score at the baseline assessment), and four were ties (i.e., no variations from the baseline to the post-assessment). As for the control group, two cases were negative, one was positive, and four were ties (Table 4). The effect sizes were reported as *r* = 0.59 for the experimental group and *r* = 0.10 for the control group. These effect sizes were transformed into *Eta^2^* = 0.35 for the experimental group and *Eta^2^* = 0.01 for the control group [45].

The CDR is scored on an ordinal scale (0–3), assessed using the Wilcoxon Signed Rank test. These results did not reveal significant differences in either group (*Z* = 0.000; *p* > 0.05). No variation was found in the CDR score, as all the comparisons were ties according to the Wilcoxon test (Table 4). The effect sizes were not computed given the small Z statistics for the comparisons in the experimental and control groups.

## 4. Discussion

This investigation aimed to explore the effects of computerized cognitive stimulation in patients with major NCDs due to AD. This study was framed as a project with SCMA that provided care to the population of a large city in the Metropolitan Area of Lisbon. The population aging is reflected in the increase of age-related diseases, where major NCDs are being considered as important challenges of developed countries for the coming years. Portugal is no exception, where there is an urgent need for the development of nonpharmacological approaches for improving the quality of life and promoting the functionality of these patients. Therefore, we developed an intervention for patients with AD built on prior work at the level of the cognitive stimulation of healthy older adults. Previous studies have shown generalized effects in cognition and executive functions (e.g., [19,22,27]). Therefore, we aimed to explore the effects of this intervention in these patients to understand whether it is possible to improve cognition in AD using an ecologically oriented approach to daily living activities in virtual reality. This function-led approach may provide advantages over paper-and-pencil exercises by focusing the intervention on functional behaviors while extending the transfer of gains to everyday living behaviors.

The results did not show improvements in the executive functions, but a significant effect was found in the global cognition between the changes in the pre and post-treatment assessments. This effect, measured by the benchmark of Cohen [44], was considered to be large. However, the differences between the groups were marginally significant at the endpoint assessment, which highlights the need for further studies in this field.

Nevertheless, the improvement in the experiment group in global cognitive functioning is aligned with the literature on cognitive stimulation in dementia that, in the absence of gains in specific cognitive dimensions, suggests improvements at the global level [15]. On the other hand, it also supports the results of studies on rehabilitation and cognitive stimulation using VR [19,22].

The outcomes on the executive functions from the TMT revealed that four patients who received interventions showed positive differences, improving from the baseline to the post-treatment assessment, and six showed no variations in results between the baseline and post-treatment in TMT part A. These results are different from those indicated by other studies (e.g., [22,46]), which points to improvements at the level of TMT part B and not at TMT part A—i.e., gains in working memory [46] but not in attention [22], motor coordination, and information processing speed [46]. However, it is not to be disregarded that, in the present study, two subjects from the experimental group revealed positive changes also in TMT part B—that is, in terms of working memory, executive functions, and the ability to switch between stimuli.

The cognitive reserve (CR) theory has been used to explain individual differences in one’s capacity to maintain cognitive function, despite the emergence of brain pathology and individual differences in the pathology [6]. For Mondini and colleagues [47], CR is a potential mechanism to cope with brain damage and, thus, facilitate cognitive performance in an impaired brain, promoting neuroplasticity mechanisms and brain reorganization following adverse events. Therefore, the CR hypothesis suggests that individuals differ in their ability to cope with AD and, further, predicts that people with a greater CR cope with advancing AD longer before the disease is observed clinically. CR is a complex construct that is viewed as an active mechanism in association with the brain´s potential to change enabled by neuroplasticity, which bears clear implications in the context of cognitive intervention [47,48].

As stated above, the CR hypothesis suggests that individuals differ in their ability to cope with the impact of AD at the level of executive and cognitive functions. CR plays an active role in the brain’s potential to change and to respond positively to CS, therefore explaining the outcome of cognitive interventions. Despite the difficulties in measuring CR, this variable would be helpful to differentiate individual performances and contribute with an explanation for intra- and interindividual differences. It is important that future studies in this topic control for CR to provide a more comprehensive understanding of the impact of CS in executive and cognitive outcomes. Hence, an effort has to be made in order to include the existing measures—the Cognitive Reserve Scale (CRS), Cognitive Reserve Index Questionnaire (CRIq), and Cognitive Reserve Questionnaire (CRQ) [49]—either in assessment protocols or in intervention plans.

Several studies have also indicated that the use of VR may be beneficial for improving the psychological functions of individuals with cognitive impairment [19,22]. However, in our study, no positive effects were found at the level of self-reported depression, as measured with the GDS-15. It is also important that further studies assess the other domains of psychological functions in terms of the quality of life and well-being, which may be important outcomes of cognitive interventions in AD.

In summary, the results indicate that VR-based cognitive stimulation can have positive effects on global cognitive functions in individuals with dementia due to AD, even though no significant changes were found in the specific cognitive functions. It is worth mentioning the difficulty in making adequate comparisons with other studies, given the fact that very few focused exclusively on the use of VR-based cognitive interventions with people with AD (e.g., [27,50]). Most of those studies integrated mild cognitive impairment (MCI) or dementia without discriminating the results related to AD (e.g., [22]), which may justify some of the differences between the results of the present study and those found in other studies. Regarding the level of immersion in VR, most studies use semi-immersive VR and, to a lesser extent, fully immersive VR [19], but a low proportion use non-immersive VR. In fact, researchers are currently debating whether fully immersive VR is better than moderate VR technology, stressing the relevance of immersion as it enhances participants’ experiences and promotes a greater sense of presence and involvement in cognitive tasks. Thus, the fact that the SLB was used for the purpose of this study in a low-immersion setup, according to the criteria of Miller and colleagues [51] on the immersion levels, may also explain the lack of significant results at the level of the primary outcome.

Moreover, another limitation was related to the lack of an active control group, i.e., a group of participants with characteristics similar to the experimental group but which were provided with a “placebo” program of activities [52,53]. The inclusion of this control group would allow us to distinguish with a greater degree of certainty whether the results obtained were due to the effects of the active component included in the intervention or whether they were only due to the placebo effects of the interventions. Psychological interventions comparatively—for example, compared to pharmacological interventions—face increasing difficulties in assessing placebo effects, since participants can usually understand what type of treatment they are receiving. Besides this more general difficulty, the specific characteristics of our experimental group did not allow us to access a third group of participants with similar and comparable characteristics. Another challenge in preparing an active control group for such interventions is the set of activities or tasks proposed for this group that were easily perceived by the participants as likely to improve the measures of the expected results to control for placebo effects [53]. Given this set of constraints, in this study, it was not possible to include an active control “placebo” group, which is an important limitation to these results. However, the inclusion of a placebo control group is an important goal for testing this approach in future definitive trials. It is therefore important that further research addresses the limitations of this study by including an active control group and controlling for potential confounders of interventions in larger samples of older adults with major NCDs due to AD in order to understand whether it is possible to delay the progression of cognitive deficits of AD with ecologically oriented VR cognitive interventions.

## 5. Conclusions

Neurocognitive disorders can have a significant impact on subjects’ quality of life, especially AD. People with AD, due to the progressive nature of the disease, end up having a global cognitive impairment and an absolute loss of autonomy. Thus, due to the severity of the impact of AD, the present study, even in the absence of an active control group—whose results indicate the potential positive effects of VR-based cognitive stimulation (using SLB)—intends to be a contribution to the literature by describing a potential approach for improving cognition in people with AD. The representativeness of the tasks used in this intervention may have also contributed to increasing patients’ retention within the treatment. The retention in this intervention program was considered high, as only one dropout was observed throughout this intervention. It is possible that the use of serious games elements (e.g., difficulty levels) contextualized in IADL contributed to improving their motivation and to retaining patients during this intervention program.

## Figures and Tables

**Figure 1 ijerph-18-05290-f001:**
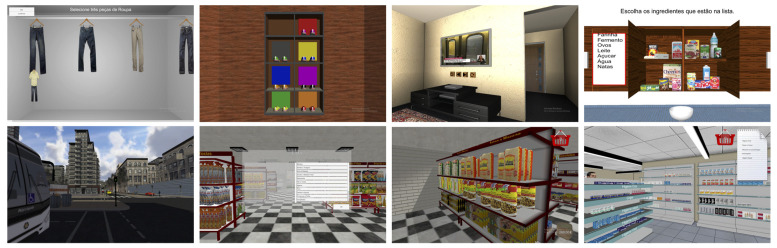
Examples of the tasks used in the Systemic Lisbon Battery

**Table 1 ijerph-18-05290-t001:** Distribution of the sociodemographic variables for the experimental and control groups.

Sociodemographic Variables	Experimental Group	Control Group
	*n*	*n*
Gender		
1.Male	3	2
2.Female	7	5
Education		
1.Below than primary school	2	0
2.Primary school	6	5
3.Higher than primary school	2	2
Civil status		
1.Married	2	0
2.Divorced	1	3
3.Widowed	7	4

**Table 2 ijerph-18-05290-t002:** Clinical Dementia Ratings at the baseline assessment.

Clinical Dementia Rating	Experimental Group			Control Group		
	Mode	Min.	Max.	Mode	Min.	Max.
Memory	1	0.5	2	1	1	2
Orientation	0.5	0.5	3	2	0.5	2
Judgment and problem solving	2	0.5	3	1	1	3
Community activities	2	0.5	2	2	1	2
Home activities	1	0.5	2	1	1	2
Personal care	1	0	2	2	0	2

**Table 3 ijerph-18-05290-t003:** Pre-post-comparisons for the parametric tests.

Outcomes	Experimental Group				Control Group				
	Baseline		Post-Test		Baseline		Post-Test		
	*M*	*SD*	*M*	*SD*	*M*	*SD*	*M*	*SD*	*F*
FAB	9.30	4.64	10.00	4.989	8.00	5.292	7.71	4.821	2.032
MMSE	18.60	6.484	19.80	7.269	13.00	7.528	12.43	7.185	4.930 *
IADL	17.20	4.050	16.60	5.190	10.71	3.861	10.29	2.984	0.015

FAB—Frontal Assessment Battery; MMSE—Mini-Mental State Examination; IADL—Instrumental Activities of Daily Living. * *p* < 0.05.

**Table 4 ijerph-18-05290-t004:** Pre-post comparisons for the nonparametric tests.

Outcomes	Experimental Group_1_			Control Group_2_			*Z_1_*	*Z_2_*
	(+) Change	(−) Change	(0) Change	(+) Change	(−) Change	(0) Change		
TMT-A	4	0	6	0	0	7	0.063	1.000
TMT-B	2	0	8	0	0	7	0.500	1.000
CDT	1	1	8	0	0	7	0.000	0.000
GDS-15	1	5	4	1	2	4	−1.897	−0.272
CDR	0	0	10	0	0	7	0.000	0.000

TMT-A/B—Trail Making Test parts A/B; CDT—Clock Drawing Test; GDS15—Geriatric Depression Scale; CDR—Clinical Dementia Rating. (+) change—*n* increase at the post-test; (−) change—*n* decrease at the post-test. In the Sign Test for the TMT, the values are *p* (probabilities), as this test computes the *p*-value directly based on the observed test statistics.

## Data Availability

The data used for this study is available upon request.

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
