# Peer review of "Virtual Reality-Based Cognitive Stimulation on People with Mild to Moderate Dementia due to Alzheimer’s Disease: A Pilot Randomized Controlled Trial"

_ijerph, 2021, doi:10.3390/ijerph18105290_

Round 1

Reviewer 1 Report

This submission is well written and the issue tackled is both timely and likely to be  broad interest. The absence of positive findings combined with the low sample size and the lack of a placebo-matched control limit the usefulness of these findings. I recognise the importance of reporting largely null effects to avoid creating biases in the literature. But it is important that studies producing largely null effects are sufficiently well designed to be able to capture positive effects. 

The following are my main concerns:

  1. Sample sizes are small. No rationale is given for the choice of sample size or reasons to assume that it is sufficiently well powered to detect differences (or power size calculations). This is an important issue that should be corrected.
  2. The section describing the matching of participants across control and experimental groups is difficult to follow. This might benefit from being in table rather than a chunk of text. Chi-square tests are used to assess differences - the statistical effects should be reported in full (and putting these in the table of matched characteristics would be helpful). In assessing matching in terms of dementia stage, I couldn't follow how this was done and I have concerns over the appropriateness of applying chi-square in this instance. For example, it should be used to assess frequencies of membership of different categories (and I couldn't understand if that was the case). It's also problematic if scores are less that 5 in one or more cells, which appears to be the case. Matching the participants across groups is important and should be clarified.
  3. Comparing an intervention against no intervention (treatment as usual) is weak as the intervention group may show placebo. I'm extremely surprised that the researchers didn't include a placebo control group of intervention that would not be expected to show benefits. To be frank, this would be reason enough to recommend rejection but I would be less concerned if this was very clearly flagged throughout the discussion and the authors were clear about the need for follow-up research to establish effects against an appropriate placebo condition control.
  4. The assessment of primarily outcome measures all seems a bit problematic. There are multiple comparisons but no apparent correction for this. At the very least the authors should report how many significant differences might be expected at chance level. These scores should all be tabulated so that they can be inspected easily. 
  5. The secondary outcome measures were a little clearer. There are some minor typographic problems in the document relating to reporting partial eta and eta values. There seems to be one positive effect. I don't think that the authors overplay its importance but I do think it's important to consider this is relation to the lack of a placebo controlled condition. 

Author Response

This submission is well written and the issue tackled is both timely and likely to be  broad interest. The absence of positive findings combined with the low sample size and the lack of a placebo-matched control limit the usefulness of these findings. I recognise the importance of reporting largely null effects to avoid creating biases in the literature. But it is important that studies producing largely null effects are sufficiently well designed to be able to capture positive effects. 

Authors’ Responses (AR): We would like to thank thoughtful the reviewer for this feedback and the opportunity to improve our study. The modifications in the revised version are highlighted in yellow text.

The following are my main concerns:

1. Sample sizes are small. No rationale is given for the choice of sample size or reasons to assume that it is sufficiently well powered to detect differences (or power size calculations). This is an important issue that should be corrected.

AR: The a priori power analysis was done using G*Power v.3.1 according to Cohen’s f effect sizes for F tests. The effect size was set to .25 (medium effect) given the moderate effect sizes obtained from previous results of virtual reality interventions on cognition of patients with dementia. Based on a power of .80 for a significance level of .05 given a medium effect size and assuming a moderate correlation among repeated measures, a total sample size of 28 patients was initially required for this trial. However, due to COVID-19 outbreak, the access to this population was interrupted from March 2020 not being possible to continue the intervention in site as originally planned. Therefore, with no perspective of returning to patient recruitment, we decided to prepare this study reporting a pilot randomized trial to provide specific effect sizes from this approach to power a future trial on this topic. We emphasized this change in the title of this revised version, describing this study as a pilot randomized controlled trial.

2. The section describing the matching of participants across control and experimental groups is difficult to follow. This might benefit from being in table rather than a chunk of text. Chi-square tests are used to assess differences - the statistical effects should be reported in full (and putting these in the table of matched characteristics would be helpful). In assessing matching in terms of dementia stage, I couldn't follow how this was done and I have concerns over the appropriateness of applying chi-square in this instance. For example, it should be used to assess frequencies of membership of different categories (and I couldn't understand if that was the case). It's also problematic if scores are less that 5 in one or more cells, which appears to be the case. Matching the participants across groups is important and should be clarified.

AR: Given the small sample size, the Fisher’s Exact statistic may be an alternative when the assumptions of the Chi-square test are not met. The associations between categorical data were all revised and replaced by Fisher’s exact test. A new table (Table 1) was also prepared to depict these data in full. Regarding the analysis on dementia classification, we have also clarified that this analysis was done according to the frequencies of membership of Clinical Dementia Rating categories.

3. Comparing an intervention against no intervention (treatment as usual) is weak as the intervention group may show placebo. I'm extremely surprised that the researchers didn't include a placebo control group of intervention that would not be expected to show benefits. To be frank, this would be reason enough to recommend rejection but I would be less concerned if this was very clearly flagged throughout the discussion and the authors were clear about the need for follow-up research to establish effects against an appropriate placebo condition control.

AR: We thank the reviewer for highlight this issue and the possibility to address this limitation. We acknowledge that this is an important issue, but it was very difficult to conduct this study including a placebo control group for testing the active component of this intervention. We will take into consideration this suggestion for the following stage of this project. We also agree that this is an important limitation to these results, and we discuss further this issue in the Discussion section.

4. The assessment of primarily outcome measures all seems a bit problematic. There are multiple comparisons but no apparent correction for this. At the very least the authors should report how many significant differences might be expected at chance level. These scores should all be tabulated so that they can be inspected easily.

AR: We have used Bonferroni correction method (alpha/m) for adjusting the alpha level in these comparisons.

5. The secondary outcome measures were a little clearer. There are some minor typographic problems in the document relating to reporting partial eta and eta values. There seems to be one positive effect. I don't think that the authors overplay its importance but I do think it's important to consider this is relation to the lack of a placebo controlled condition.

AR: The typos in reporting the effect measures are now corrected. We have also calculated the effect size for all statistical procedures.

Reviewer 2 Report

Dear Authors, 

Thank you for the opportunity to review the paper entitled “Virtual Reality-Based Cognitive Stimulation on People with Mild to Moderate Dementia due to Alzheimer’s Disease: A Randomized Controlled Trial”. The paper deals with a very interesting topic, not fully explored. In my opinion, the paper contributes valuable information in an area that will grow rapidly in the future. The paper has a clear purpose, the clarity of the manuscript is high, with good scientific soundness.  However, I miss a bit of information in the discussion of VR effects at the physiological level. The issue of cognitive reserve is hinted upon, but the authors are rather timid in their interpretation of their results. I would encourage you to elaborate on why there would be differences between traditional training and VRET?

Minor issues:

1. 2.7 Statistical Analysis

  • Please provide details on the SPSS manufacturer (company, city, country)
  • Please indicate a description of the sample size calculation
  • Please also indicate in this section that the effect size was calculated.

2. I miss a separate conclusion section

3. Lack of information on the clinical trial registration

Author Response

Thank you for the opportunity to review the paper entitled “Virtual Reality-Based Cognitive Stimulation on People with Mild to Moderate Dementia due to Alzheimer’s Disease: A Randomized Controlled Trial”. The paper deals with a very interesting topic, not fully explored. In my opinion, the paper contributes valuable information in an area that will grow rapidly in the future. The paper has a clear purpose, the clarity of the manuscript is high, with good scientific soundness.  However, I miss a bit of information in the discussion of VR effects at the physiological level. The issue of cognitive reserve is hinted upon, but the authors are rather timid in their interpretation of their results. I would encourage you to elaborate on why there would be differences between traditional training and VRET?

Author Responses (AR): We would like to thank thoughtful the reviewer for this feedback and the opportunity to improve our study. We have discussed further our results to explain the active component of this approach for improving cognition. The modifications in the revised version are highlighted in yellow text.

Minor issues:

  1. 2.7 Statistical Analysis
  • Please provide details on the SPSS manufacturer (company, city, country)

AR: The details for this software were included in this revised version of the manuscript.

  • Please indicate a description of the sample size calculation
  • Please also indicate in this section that the effect size was calculated.

 AR: The a priori power analysis was done using G*Power v.3.1 according to Cohen’s f effect sizes for F tests. The effect size was set to .25 (medium effect) given the moderate effect sizes obtained from previous results of virtual reality interventions on cognition of patients with dementia. Based on a power of .80 for a significance level of .05 given a medium effect size and assuming a moderate correlation among repeated measures, a total sample size of 28 patients was initially required for this trial. However, due to COVID-19 outbreak, the access to this population was interrupted from March 2020 not being possible to continue the intervention in site as originally planned. Therefore, with no perspective of returning to patient recruitment, we decided to prepare this study reporting a pilot randomized trial to provide specific effect sizes from this approach to power a future trial on this topic. We emphasized this change in the title of this revised version, describing this study as a pilot randomized controlled trial. We have also calculated the effect size for all statistical procedures.

  1. I miss a separate conclusion section

AR: The conclusion is now presented in a separate section.

  1. Lack of information on the clinical trial registration

AR: This trial was not registered in a public database. However, if the reviewer/editorial team agree, we may register this trial retrospectively in a public database (e.g., clinicaltrials.gov). This may also be justified because this study is part of an ongoing project for developing an ecologically-oriented approach for improving cognition in Alzheimer’s disease.